# Insights into the Multidisciplinary Approach for Metastatic Acinic Cell Lung Carcinoma: The Pathologist’s Role in Romania Today

**DOI:** 10.3390/curroncol32010037

**Published:** 2025-01-13

**Authors:** Dorela-Codruta Lazureanu, Amelia Burlea, Robert Barna, Daniela Cipu, Mihaela Pasca Fenesan, Ioan Icma, Marioara Cornianu

**Affiliations:** 1Microscopic Morphology Department—Anatomical Pathology, ANAPATMOL Research Center, “Victor Babes” University of Medicine and Pharmacy, 300041 Timișoara, Romania; lazureanu.dorela@umft.ro (D.-C.L.); robert.barna@umft.ro (R.B.); cornianu.marioara@umft.ro (M.C.); 2Pathology Department, “Pius Brinzeu” Emergency County Clinical Hospital Timisoara, 300723 Timisoara, Romania; 3Pathology Department, Emergency County Hospital Arad, 310037 Arad, Romania; ameliaburlea@gmail.com; 4Department of Pathology, Center for Research and Innovation in Precision Medicine of Respiratory Diseases, “Victor Babes” University of Medicine and Pharmacy, 300041 Timisoara, Romania; 5Radiology and Medical Imaging, “Victor Babes” University of Medicine and Pharmacy, 300041 Timișoara, Romania; 6“Pius Brinzeu” Emergency County Clinical Hospital Timisoara, 300723 Timisoara, Romania; 7OncoHelp Hospital, 300239 Timisoara, Romania; mihaela.fenesan@umft.ro; 8Department of Microscopic Morphology/Histology, “Victor Babes” University of Medicine and Pharmacy, 300041 Timisoara, Romania; 9Surgery Department Ist Clinic, “Pius Brinzeu” Emergency County Clinical Hospital Timisoara, 300723 Timisoara, Romania; nutu75@yahoo.com

**Keywords:** acinic cell carcinoma, lung, metastases, ancillary tests, multidisciplinary approach

## Abstract

**Background/Objectives**: Acinic cell carcinoma (ACC) is a rare lung neoplasm that can affect both children and adults as a parenchymal or endobronchial mass. It is histologically similar to this kind of tumor described in salivary glands, but with a different immunophenotype. In general, it poses a reduced degree of malignancy, with indolent growth and a favorable prognosis, with exceptionally rare cases associated with recurring disease or lymph node metastases. **Methods**: When clinicians are facing puzzling symptomatology in their patients, the main role of the multidisciplinary team in their review of oncological cases is to recommend imagistic-guided biopsies. Tissues samples were routinely processed, stained with hematoxylin-eosin (HE) and periodic acid–Schiff (PAS), and submitted to complementary immunohistochemistry tests. **Results**: Histopathological reports were consistent for lung ACC with regional lymph node involvement and remote metastases. Oncological therapies followed. **Conclusions**: Postponements of the presentation to the doctor at the onset of symptoms, as well as a lack of periodic health control for people insured by national health insurance companies, often lead to medical, human and financial complications that are difficult to manage. The pathologist involved in the discussion of oncological cases brings his expertise in solving cases, certifying the evolution of tumors considered less aggressive, such as in the case of lung ACCs.

## 1. Introduction

Acinic cell carcinoma (ACC), also known as Fechner tumor, is a rare lung neoplasm, usually found in adults as a parenchymal or endobronchial mass, that is histologically similar to this kind of tumors described in salivary glands. So far, there have been only 25 cases reported in the literature, among both adults and children [1,2,3]. The first case of acinic cell pulmonary tumor was described by Fechner in 1972 [4], in a 63-year-old patient who presented a 42 mm tumor in the right lower lobe that did not involve the lobar bronchus. The lesion described, which had developed (most probably) from the serous cells of submucosal bronchial glands, was termed acinic cell tumor, similar to those found in salivary glands [5].

Since then, similar cases have been published, with central or peripheral localization in the lung. Mixed tumors, consisting of acinic cells and neuroendocrine (NE) cells, are even rarer, with the literature describing only two cases with mixed structures (carcinoid and acinic cell tumor) [6,7,8]. Patients diagnosed with this tumor fell in a broad age range, from 4 to 75 years old, with a median of 48 years old. There was no gender predisposition, and nearly half of the patients were asymptomatic, while the rest experienced coughing and hemoptysis, among the more common symptoms [3,9]. In general, these are malignant neoplasms, but they have reduced aggressiveness, which associated with indolent growth and a favorable prognosis. Recurrent disease or with lymph node metastases, have been seldom reported, and according to our knowledge, no distant metastases have been observed so far.

## 2. Materials and Methods

Within a relatively short timeframe, the MDT debate began to focus on two patients: a 57-year-old woman, and a 70-year-old male, with intriguing clinical symptomatology and complex input data from thoraco-abdominal computed tomography (CT), fiber bronchoscopy, and MRI (of the female patient) from the brain to the neck and thorax after a physical examination of the anterior aspect of the neck (the male patient). In both situations, histopathological examination was recommended.

Tissue material (from biopsies) was processed using the standard method: 10% neutral buffered formalin fixation, followed by paraffin embedding. Sections were stained with HE, PAS (with and without previous diastase digestion) and Alcian Blue. For the immunohistochemical (IHC) profile, we used antibodies anti-cytokeratin AE1/AE3, CAM 5.2, EMA, chromogranin A (CgA), synaptophysin (Syn), thyroglobulin, S100, TTF-1, and Ki-67; we performed visualization with the Ultravision system, DAB visualization and hematoxylin counterstaining.

## 3. Results

The 57-year-old female patient, a smoker, without a personal pathological history or recent respiratory symptoms, was admitted with confusion syndrome with a recent onset, marked asthenia, moderate weight loss, loss of appetite and vomiting (not preceded by nausea, but followed by accentuated headache). General clinical examination revealed the presence of warts and dysplastic nevi disseminated on the abdominal and thoracic skin. Her salivary glands, oral cavity and nasopharynx showed no significant signs.

Thoracic–abdominal computed tomography (CT) revealed the following:-A right hilar lung tumoral mass of 35 × 32 mm, with the amputation of the middle lobar bronchus and an atelectatic right–middle pulmonary lobe; the presence of a 29 × 23 mm lesion in the lower right lobe, attached to the pleural and diaphragmatic surfaces (Figure 1A);-Mediastinal and right hilar lymphadenopathies;-A 16 mm hypodense lesion in the right hepatic lobe (constrast-enhancing) (Figure 1B) and one lesion measuring 12 mm in the sixth segment.

Brain CT detected multiple lesions with a hyperdense halo at the periphery, with the largest of all measuring 20 × 19 mm with surrounding edema, located in the left frontal–parietal area, as well as other smaller lesions, with the same characteristics, involving the frontal lobes and the left parietal–occipital area. Brain MRI revealed multiple round-to-oval lesions of variable dimensions up to 24 mm, in the infra- and supratentorial areas, diffusely distributed in both hemispheres (Figure 2A), in the cerebellum hemispheres and in the right half of the brain stem, with surrounding edema; these lesions were interpreted as multiple secondary determinations, with a mass effect on the left lateral ventricle (Figure 2B).

Fiber bronchoscopy identified a proliferative process developed on the anterior wall of the medium lobar orifice, with almost complete obstruction of it; a biopsy was taken from this mass.

Microscopically, there was monomorphic tumor proliferation, with a solid growth pattern, predominantly composed of sheets and nests of round-to-oval cells, and with distinct cellular limits, a rich, clear and vacuolated cytoplasm, and relatively uniform nuclei, which were round or oval; occasionally, there were evident nucleoli, situated eccentrically, mimicking “signet ring cells”. The second type of cells had a eosinophilic granular cytoplasm (Figure 3A) and slightly hyperchromic nuclei that were smaller than those of clear cells. The mitotic activity was low, at 2–3 mitoses / 10HPF. Tumor cell nests were separated by fine fibrous bands, including blood vessels with thin walls, mostly capillaries, focally surrounded by lymphocytes. Histochemically, tumor cells were PAS-positive (Figure 3B). All cells were negative in the Alcian Blue stain.

Immunohistochemically, tumor cells were positive for AE1/AE3 cytokeratin (Figure 4A), EMA (cytoplasmic pattern), and TTF-1 (nuclear pattern) (Figure 4B), and were negative to neuroendocrine markers (CgA and Syn) and S100. The cell proliferation index using the Ki-67 antibody was about 7% (Figure 4C).

The 70-year-old male with a history of nodular goiter presented hoarseness and difficulty in breathing. Physical examination revealed a 6 × 4 cm, multinodular mass within the left thyroid lobe and enlarged lateral cervical lymph nodes. MRI with the contrast substance showed an enlarged, multinodular left thyroid lobe with heterogeneous contrast uptake, alongside confluent lateral cervical lymphadenopathies and bilateral nodular pulmonary lesions. Biopsies of the thyroid and cervical lymph nodes lesions were performed.

Histopathological examination of both thyroid gland and lateral cervical lymphadenopathy revealed a monomorphic tumoral proliferation, arranged in lobules, micropapilly and nests of atypical cells with abundant, clear (Figure 5A), and, in some areas, granular, eosinophilic cytoplasm; these cells were also diastase–PAS-positive (Figure 5B), with eccentric, vesicular nuclei and low mitotic activity. The tumor displayed a fibrous, hyalinized stroma, vascular invasion, occasional psammoma bodies, and small foci of necrosis.

Immunohistochemistry showed positivity for CAM 5.2 (Figure 6A) and TTF1 (Figure 6B), while thyroglobulin, S100 and chromogranin A were negative.

Both patients underwent chemoradiotherapeutical protocols. The outcome was unfavorable, with numerous treatment- and disease-related complications shortly after, followed by death within 3 months for the female patient and within 6 months for the male patient.

## 4. Discussion

There is no doubt that only interdisciplinary collaboration involving the exchange of essential information from each medical branch involved in the management of oncological patients can lead to personalized therapy with maximized benefits for the patient and for the national health system. And within the multidisciplinary team of the tumor board, the pathologist plays the role of revealing the therapeutic target and providing the prognostic assessment.

For the presented situations, the histopathological result was surprising, not so much as the starting point of the multisystem metastases but from the perspective of the histopathological type of malignancy in particular, because acinic cell carcinoma (ACC) is a rare tumor in the lung with a low grade of malignancy that grows slowly and, most of the time, has a favorable prognosis. It seldom metastasizes [2], with only 25 cases having been described in the literature to this date [3].

Macroscopically, acinic cell tumors are well-circumscribed lesions, but they are not encapsulated. Generally, they measure between 10 and 50 mm, their consistency is soft or elastic, and they are of a whitish-brown or yellow color, without necrosis and hemorrhage [4]. Endobronchial polypoid tumors are well-circumscribed nodules that gray–white in color and soft or elastic, covered by a smooth mucosa [1].

Histological aspects of ACC are identical to its counterpart in the salivary glands. These tumors comprise polygonal, uniform tumor cells, with clear, vacuolated cytoplasm, small round nuclei located eccentrically (signet ring cells) and discrete nucleoli, as well as cells with rich, eosinophilic, granular cytoplasm and a positive reaction to PAS [2]. The tumor has a predominantly solid pattern of growth, with cellular sheets that can be separated by thin connective bands. Besides the solid pattern, other morphological growth patterns have been described: acinic or glandular, microcystic, nesting or papillary–cystic [1]. Sometimes, the tumor can comprise cystic spaces lined by neoplastic cells. Some tumors can be predominantly made up of oncocytic cells, with a nesting pattern of growth, similarly to that of neuroendocrine tumors [2,9]. Rarely, ACC can be associated with perineural invasion [4].

Histochemical features are variable; neoplastic cells can be positive with PAS, d-PAS and mucicarmin reactions. The most important diagnostic feature is the cytoplasmic presence of zymogen dense granules that are ultrastructurally (US) evident at a 600–800 nm diameter, close to the cell membrane, similarly to those observed in serous acini of tracheobronchial glands [1,2]. Just one of the five cases reported by Moran et al. [6] reacted intensely with PAS or PAS-D, with the tumors having a lower level of amylase than normal acinic cells.

The IHC profile of ACC is defined by its intense reactivity to cytokeratins CKAE1/AE3, CAM 5.2 and EMA and by its absence of immunoreactivity to the S100 protein, chromogranin and vimentin. At the same time, its DOG-1 positivity is debatable [10]. This IHC profile can distinguish pulmonary ACC from other neoplasms, such as clear-cell or granular PAS-positive tumors, including the “sugar” tumor, oncocytic carcinoid, oncocytoma, granular cell tumor; primary and metastatic (of renal origin) clear-cell carcinoma; bronchioloalveolar carcinomas rich in glycogen; acinic cell metastatic tumors; granular cell tumors of the head and neck regions [4].

Generally, acinar cell tumors of the lung have a favorable clinical evolution, with no predisposition to metastases. After reviewing 18 cases, Ukoha et al. [11] described the involvement of an interlobar lymph node (pN1) in only one patient aged 64 years old with an asymptomatic lung tumor. Another case of primary lung acinic cell carcinoma with metastases in hilar lymph nodes was reported by Lee et al. [12] in a 30-year old woman with a subpleural tumor mass of 100 × 60 × 20 mm in the inferior lobe of the right lung, found incidentally on thoracic routine radiography. Histopathogically, the tumor mitotic activity was the same as that in cases presented (3/10 HPF) with evidence of perineural invasion. The tumor recurred 20 months after complete resection. The same behavior—recurrence after surgical removal—was reported by Chuah et al. [13].

Essential for the presented cases, in pointing out that lung masses were the origin of multiple-organ metastases, was the immunohistochemically positive reaction for TTF1 in conjunction with imaging. This association of TTF1 positivity in the neoplastic cells with the existence of the lung tumor observed by imaging is absolutely necessary to determine, knowing that other tumors with extrapulmonary site show TTF1 positivity: thyroid and CNS (diencephalus) tumors. Moreover, in males, negative complementarity thyroglobulin has been ruled out in primary thyroid carcinoma [14].

## 5. Conclusions

It remains, for Romania, necessary to implement national screening programs for the detection of malignant neoplasms early on in their evolution. Otherwise, regardless of the intensity of human and material efforts, oncologic patients diagnosed in the advanced/metastatic stage of the disease—even for tumors with mild biological behavior, such as acinar cell carcinoma—will not have much of a chance to survive.

## Figures and Tables

**Figure 1 curroncol-32-00037-f001:**
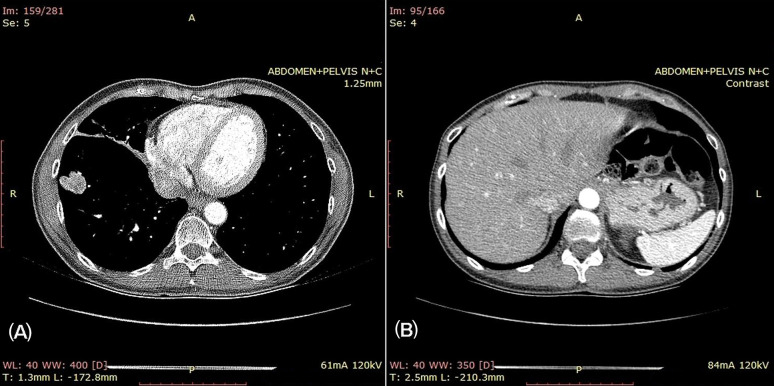
Thoracic and abdominal CT. (**A**) Tumor lesion in the inferior lobe of the right lung, touching the pleural surface; (**B**) hypodense lesion in the hepatic right lobe, retaining the contrast substance.

**Figure 2 curroncol-32-00037-f002:**
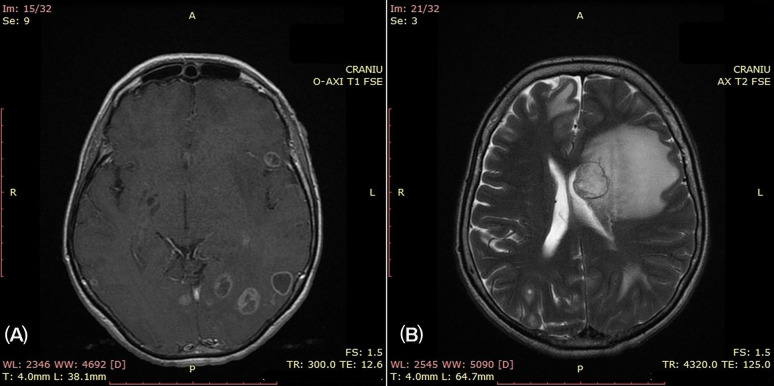
Brain MRI. (**A**) Multiple round-oval supratentorial lesions in both brain hemispheres; (**B**) marked perilesional edema with a mass effect on the left ventricle.

**Figure 3 curroncol-32-00037-f003:**
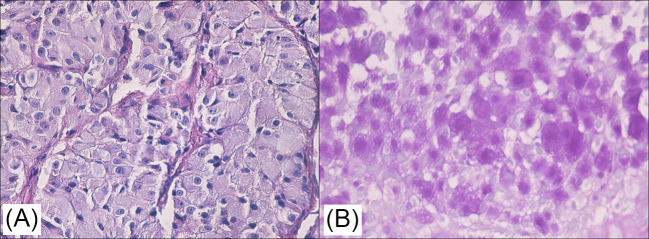
(**A**) Monomorphic tumor proliferation made up of sheets of round/oval cells, with distinct cell margins, and a rich, clear, slightly eosinophilic or/and granular cytoplasm (HE); (**B**) diastase–PAS-positive reaction in the cytoplasm of granular tumor cells (original magnification: ×400 for both images).

**Figure 4 curroncol-32-00037-f004:**
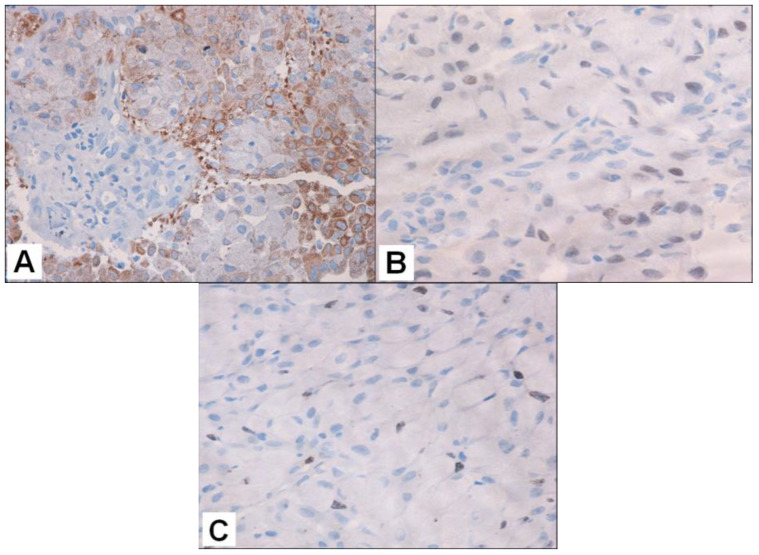
Immunohistochemical features of the tumor. (**A**) AE1/AE3 cytokeratin-positive immunoreaction, with a diffuse, granular cytoplasmic pattern (original magnification: ×200); (**B**) TTF1 and (**C**) Ki-67 positive expression, a showing nuclear pattern (original magnification: ×400 for both images).

**Figure 5 curroncol-32-00037-f005:**
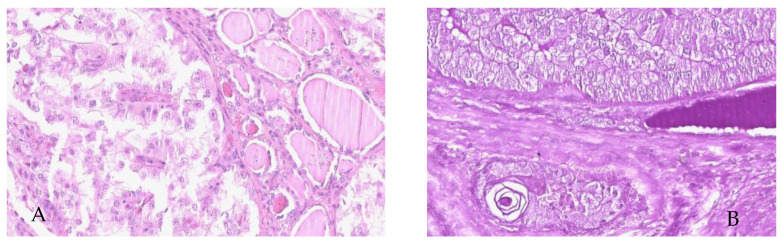
(**A**) Thyroid tumor: nests of round/oval cells, with a rich, clear, slightly eosinophilic or/and granular cytoplasm shown via HE staining; original magnification: ×200 (**B**) Diastase–PAS-positive reaction in the cytoplasm of tumor cells, the colloid of the non-neoplastic follicle and a psammoma body; original magnification: ×400.

**Figure 6 curroncol-32-00037-f006:**
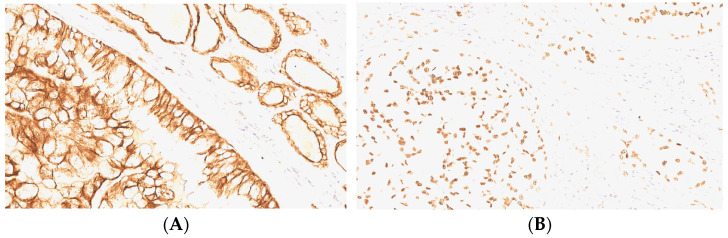
(**A**) Lower-left-side thyroid tumor positive for CAM 5.2 cytokeratin; original magnification: × 400. Same tumor positive for (**B**) TTF1; original magnification: ×200.

## Data Availability

No new data were created or analyzed in this study. Data sharing is not applicable to this article.

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
