# Peer review of "Insights into the Multidisciplinary Approach for Metastatic Acinic Cell Lung Carcinoma: The Pathologist’s Role in Romania Today"

_curroncol, 2025, doi:10.3390/curroncol32010037_

Round 1

Reviewer 1 Report

Comments and Suggestions for Authors The reviewed article consists of the following parts - introduction, material and methods, results, discussion and conclusions; contains 13 literature items. The authors present 2 cases of ACC lung cancer with multiple distant metastases. The rare nature of this tumor is noteworthy. Histopathologically, it is very similar to ACC occurring in the salivary glands, which is also not common. The topic of the work - the insight of a multidisciplinary approach in metastatic ACC: the role of a pathologist in Romania today corresponds to the content of the article.
The results chapter describes the symptoms occurring in the presented patients - according to the reviewer, this should be included in the material and methods chapter because the symptoms are difficult to call results because the patients were already admitted to the authors' center with these symptoms. Overall, the assumptions formulated for the work were implemented. The authors use correct terminology; the work meets the criteria of a scientific text. I consider the structure of the article and its linguistic correctness to be correct. In the article, the authors used more important and relatively new literature items (the latest item from 2019). More current literature items can be added.
The illustrations are selected properly and complement the text of the article correctly. The conclusion of the article is that screening tests could perhaps allow for earlier diagnosis of ACC in the described patients, which is a conclusion corresponding to the problem posed, i.e. an interdisciplinary approach in oncology facilitating the detection of cancers in earlier stages that can be treated more effectively. The authors did not present what type of radio-chemotherapy they used. Please add images of chest CT with bilateral nodular pulmonary lesions ( 70 years-old male) The reviewer believes that after taking into account the above comments, the work qualifies for publication.

Author Response

Thank you for your review and valuable suggestions.

Comment 1: In the article, the authors used more important and relatively new literature items (the latest item from 2019). More current literature items can be added.

Answer DCL: Regarding the references, the 2019 reference was the most recent one we could find related to lung acinic cell carcinoma with metastases. We acknowledge the existence of newer references, such as this article from 2021 (https://bmcsurg.biomedcentral.com/articles/10.1186/s12893-021-01351-8) and this abstract from 2021 (https://www.jto.org/article/S1556-0864(21)02848-3/fulltext). However, these studies discuss tumors without metastases or recurrence, which fall outside the scope of our paper. If you consider them relevant, we are happy to include them. Alternatively, if you can suggest other more suitable recent references, we would be happy to cite them.

Comment 2: The authors did not present what type of radio-chemotherapy they used.

Answer DCL: unfortunately, the medical records we accessed only indicate that such therapy was administered. We do not have detailed information about the specific regimens.

Comment 3: Please add images of chest CT with bilateral nodular pulmonary lesions (70 years-old male)

Answer DCL: Similarly, while we have the radiology report from the chest CT, we do not have access to the original images and therefore cannot provide further details on this aspect.

Reviewer 2 Report

Comments and Suggestions for Authors

In this paper, the authors describe acinic cell tumors in two patients, one with pulmonary Fechner's tumor. Interestingly, the cases were discovered at locally extensive and metastatic stages, respectively, and led to death rapidly after histological diagnosis in both cases, despite the treatments administered.

The manuscript is generally well-written; however, certain parts, such as the treatment regimens and the reasons for the lack of molecular pathological tests, require clarification.

The pathological diagnosis of the tumors is nicely documented; the cancer was confirmed by immunohistochemistry in both cases. However, either patient's molecular pathological tests (e.g., PD-L1 status) are unavailable. Was there a specific reason for this not being reported? The references are appropriate, but I would suggest some additions to the literature, e.g., Nibid et al. (2022), Yang et al. (2023), and Huang et al. (2024), as these studies provide further insights into acinic cell tumors and could enhance the discussion section of your manuscript.

Author Response

Thank you for your helpful feedback and valuable suggestions. We appreciate your effort in improving the quality of our manuscript.

Comment 1: However, either patient's molecular pathological tests (e.g., PD-L1 status) are unavailable. Was there a specific reason for this not being reported?

Answer DCL: We gathered all available information from the medical records but regretfully do not have access to additional data. In terms of molecular testing, the patients did not have access to supplementary tests, and we no longer have the FFPE tissue available to perform further analyses.

Comment 2: The references are appropriate, but I would suggest some additions to the literature, e.g., Nibid et al. (2022), Yang et al. (2023), and Huang et al. (2024), as these studies provide further insights into acinic cell tumors and could enhance the discussion section of your manuscript.

Answer DCL: We have included the article by Nibid et al. in our revised manuscript, but we could not locate the Yang et al. reference. Regarding the Huang et al. study, since it reports on a case of ACC without metastases or recurrence, which differs from our focus, we did not include it in the current version of the manuscript. However, if you believe it would be useful, we are happy to incorporate it.